# Impact of Maternal Age on Singleton Pregnancy Outcomes in Primiparous Women in South Korea

**DOI:** 10.3390/jcm11040969

**Published:** 2022-02-12

**Authors:** Eui Hyeok Kim, Jaekyung Lee, Sang Ah Lee, Yong Wook Jung

**Affiliations:** 1Department of Obstetrics and Gynecology, National Health Insurance Service Ilsan Hospital, Goyang-si 10444, Korea; raksumi10@gmail.com; 2Department of Obstetrics and Gynecology, Severance Hospital, Yonsei College of Medicine, Seoul 03722, Korea; jjackielee@yuhs.ac; 3Big DATA Strategy Department, National Health Insurance Service, Wonju-si 26464, Korea; laurenleey91@gmail.com; 4Department of Obstetrics and Gynecology, CHA Gangnam Medical Center, CHA University School of Medicine, Seoul 06135, Korea

**Keywords:** gravidity, maternal age, pregnancy complications, risk factors, cesarean section

## Abstract

We aimed to determine the association between maternal age and pregnancy outcomes in singleton primigravidae in South Korea. We reviewed the medical data of singleton primigravidae women who gave birth between 2013 and 2018 and underwent the National Health Screening Examination using the administrative database from the National Health Insurance claims data. As advanced maternal age is associated with various comorbidities that could affect pregnancy outcomes, we divided the patients according to their age and comparatively analyzed the prevalence of high-risk pregnancy complications including rates of cesarean delivery, after adjusting for maternal demographics. Perinatal and postpartum complications according to maternal age were also investigated. Overall, 548,080 women were included in this study: 441,902 were aged <35 years; 85,663, 35–39 years; 20,515, ≥40 years. Patients’ demographics differed according to their age. Increasing maternal age was significantly associated with higher income levels and higher rates of obesity, pre-existing diabetes, and hypertension. With the increasing maternal age, the rate of obstetric complications, including gestational diabetes, preeclampsia, placenta previa, placental abruption, and cesarean delivery, increased. Maternal age was also positively correlated with perinatal morbidity including preterm birth and low birth weight. Additionally, advanced maternal age was a risk factor for hospitalization before delivery, more frequent outpatient visits, and readmission after delivery. These observations were maintained in the multivariate analysis results. Advanced maternal age appears to be associated with various adverse obstetric outcomes for primigravidae women, and the frequency of hospitalizations was higher in this group. Considering the current social environment of late pregnancies and childbirth in South Korea, effective policy consideration is required to support safe childbirth in women with advanced maternal age.

## 1. Introduction

Advanced maternal age (AMA) is defined as an age ≥35 years on the estimated date of delivery. There is a notable trend towards increasing maternal age at childbirth worldwide, particularly in developed countries. Data from The Organization for Economic Cooperation and Development revealed that most women in developed countries currently have their first birth at ≥30 years of age; this age has risen by approximately 5 years over the last two decades globally [1]. In the United States, 9% of all the first births in 2014 were to mothers of AMA, and this represented a 23% increase from 2000 [2]. Furthermore, the mean age at first birth in the United States has also increased by approximately 5 years (from 21.4 to 26.3 years) between 1970 and 2014 [2].

South Korean data indicate that the total number of births has reduced by 51.2% over the last two decades (from 620,668 in 1999 to 302,676 in 2019). However, the proportion of women of AMA has increased from 6.3% in 1999 to 33.3% in 2019 over the last two decades [3]. South Korea ranked first in terms of the number of mothers of AMA at first birth (mean age, 31.6 years) [3]. 

Although some studies have observed that most women >45 years of age have good pregnancy outcomes and can cope with the physical and emotional stress of pregnancy and parenting [4,5,6,7,8], there have been numerous studies regarding various adverse maternal outcomes associated with AMA, including fetal chromosomal abnormalities, cesarean delivery, preeclampsia, postpartum hemorrhage, gestational diabetes mellitus (GDM)/diabetes mellitus (DM), thrombosis, hysterectomy, and, in adjusted analyses, amniotic fluid embolism and increased risk of obstetric shock [9,10,11,12]. In addition, unfavorable infant outcomes, such as preterm birth, low birth weight (LBW), small for gestational age, and perinatal mortality, could be more likely in an AMA group [13,14,15].

There are few studies on the relationship between maternal age and pregnancy outcomes based on a South Korean population. Previous studies on women of AMA have been performed either with relatively small samples or without adequate pre-pregnancy demographic data of the patients. Older women are generally more likely to have pre-existing comorbidities, such as obesity, DM, and hypertension [11]. With these risk factors, we can expect women younger than 35 years of age to have higher rates of hospitalization, cesarean delivery, and pregnancy-related complications than their healthy counterparts. 

It is important to know how patient characteristics affect pregnancy outcomes according to maternal age. Counseling and education are needed for more informed decision making on pregnancy at an AMA. Big data platforms are changing biomedical research with large-scale, data-driven research. Recently, the South Korean National Health Insurance Database has provided researchers open access to data for public research purposes. The present nationwide South Korean study will provide actual, up-to-date insights into the impact of AMA on pregnancy outcomes. This study evaluated the overall pre-pregnancy risk factors for adverse pregnancy complications in women of AMA and sought to determine the impact of maternal age on pregnancy outcomes in singleton primigravidae in South Korea. 

## 2. Materials and Methods

### 2.1. Study Population

In South Korea, about 97% of the population is obligated to enroll in the National Health Insurance Service (NHIS), while the remaining 3% are covered by medical care to protect them against the financial burden of excessive medical expenditures. Therefore, the NHIS database contains information on all the claims for approximately 50 million South Koreans, including both types of beneficiaries. Most of the information about the extent of a disease can be obtained from this centralized database. Data on the economic status of the beneficiaries, divided into 20 grades, can also be obtained from the NHIS database. Additionally, as part of the NHIS healthcare program, beneficiaries who satisfy specific criteria (self-employed beneficiaries who are insured, beneficiaries of NHI employee insurance, and dependent family members of beneficiaries of NHI employee insurance) are invited to participate in a biannual National Health Screening Examination (NHSE). These screening results are recorded in the NHIS database. 

The present study used customized health data from the NHIS database that can be provided on reasonable request, in accordance with the policy on academic research (https://nhiss.nhis.or.kr). Using this database, we identified all women who had their first delivery with a singleton pregnancy between 1 January 2013 and 31 December 2018 in South Korea, using the diagnosis based on the 10th revision codes of the International Classification of Disease and the relevant procedure codes.

To facilitate the evaluation of pre-pregnancy characteristics, only women who underwent an NHSE at least 2 years before their first delivery were included. The NHSE consists of a health interview and health examination. The health interview includes questions regarding socioeconomic, demographic, and lifestyle status, including exercise, smoking, and alcohol consumption. Women with multiple pregnancies, those on medical aid, and multiparous women were excluded from this study. A total of 548,080 women who experienced their first delivery with a singleton pregnancy between 2013 and 2018 were included in this study.

This study was approved by the ethics committee of the National Health Insurance Service Ilsan Hospital (approval number: NHIMC 2020-06-007). The requirement of informed consent was waived due to the study’s retrospective and observational design.

### 2.2. Identification of Outcomes and Pre-Pregnancy Factors

The primary outcomes of this study were late pregnancy complications including GDM, preeclampsia, placental abruption, placenta previa, and need for a cesarean section. The secondary outcomes were perinatal morbidity (including low birth weight, preterm delivery, and stillbirth/intrauterine fetal death [IUFD]), and postpartum complications. The postpartum complications were evaluated based on postpartum hemorrhage requiring transfusion, readmission rate within 30 days of discharge, and the number of outpatient visits within 50 days of delivery. We measured these adverse pregnancy outcomes based on the medications used because of the limitations of the NHIS claims database. For example, GDM/DM was assessed based on insulin treatment, and preeclampsia was assessed based on magnesium sulfate treatment. LBW was defined as a birth weight of less than 2500 g, regardless of the gestational age of the infant. For LBW, the ICD-10 codes P07.0 and P07.1 were used. Preterm birth was defined as delivery before 37 gestational weeks (ICD-10 of O60.1 and O60.3). For GDM, DM, placenta previa, placental abruption, stillbirth, and IUFD, the ICD-10 codes O24.4, E1, O44, O45, P95, and O36.4 were used, respectively. The ICD-10 codes of O11, O14, and O15 were used for preeclampsia. We also included the pre-pregnancy factors from the NHSE data to adjust for the differences in women with first delivery in this study and considered the following variables: age, body mass index (BMI), exercise, smoking, and alcohol consumption. BMI was divided into low (<18 kg/m^2^), normal (18.5–22.9 kg/m^2^), overweight (23–24.9 kg/m^2^), and obesity (≥25 kg/m^2^). These values were adopted from the cutoffs established for the proposed classification of weight by BMI in adult Asians [16]. Economic statuses were divided into quartiles according to the recorded income status data in the NHIS: low, low-middle, middle-high, and high. The included women were categorized into non-heavy or heavy drinkers based on their alcohol consumption habit. The Ministry of Health and Welfare (MOHW) in South Korea defined heavy drinking as an alcohol consumption status that needed correction, i.e., drinking alcohol more than four times per week or taking more than four drinks at a time. This definition is based on the National Institute on Alcohol Abuse and Alcoholism (NIAAA) and revised by the MOHW in consideration of the South Korean population. The included women were categorized into current (at the time of NHSE health interview) or non-smokers based on their pre-pregnancy smoking status. The US Department of Health and Human Services published physical activity guidelines for Americans to promote health and reduce the burden of chronic disease. Based on this guideline, the MOHW produced the physical activity guide for Koreans. According to this activity guide, we categorized the subjects into two groups: physically active and inactive. Physically active was defined as more than three episodes of high-intensity workouts per week or as more than five episodes of intermediate workouts per week. The residential area was divided into three categories: Capital (Seoul and Gyonggi), Metropolitan area (Incheon, Daejeon, Gwangju, Busan, Daegu, and Ulsan), and others. 

Charlson Comorbidity Index (CCI) is the measure of the comorbid disease status. A weighted score was assigned to each of the 17 comorbidities, based on 1-year mortality. The sum of the index score is an indicator of disease burden and a strong estimator of mortality. We calculated the index using ICD-10 [17]. 

### 2.3. Statistical Analyses

We first examined the general characteristics of the study population. Continuous and categorical variables were expressed as mean and frequency, respectively. In addition, we performed Student’s t-tests, chi-square tests, or analysis of variance to identify the differences in the distribution of general characteristics and complications according to age group. Finally, we performed Poisson regression, linear regression, and logistic regression analyses to estimate the adjusted odds ratio (aOR) and 95% confidence interval (CIs) for risk diagnosis. All statistical analyses were performed using SAS version 9.4 (SAS Institute, Inc.; Cary, NC, USA). A *p*-value of <0.05 was considered statistically significant.

## 3. Results

A total of 2,354,129 women gave birth between 2013 and 2018 in South Korea and 861,047 of these women had undergone NHSE at least 2 years before delivery. The NHIS data of 548,080 women were gathered based on the inclusion criteria described above: nulliparous women with singleton deliveries between 2013 and 2018. The patients were subdivided into three groups according to maternal age: (1) 441,902 (80.6%) women were <35 years old; (2) 85,663 (15.6%) women were 35–39 years old; (3) 20,515 (3.7%) were ≥40 years of age (Figure 1). 

Patients’ characteristics, except for age, were collected before their pregnancy. The data are shown in Table 1. 

Older pregnant women were more likely to be in the high-income group (*p* < 0.001) and the capital/metropolitan resident group (*p* < 0.001). Furthermore, a higher proportion of smokers were found among older patients (*p* < 0.001); however, they were less likely to drink (*p* < 0.001). Regarding comorbidities, increasing maternal age was associated with higher rates of obesity (*p* < 0.001), pre-existing DM (*p* < 0.001), and hypertension (*p* < 0.001). The CCI also showed a tendency towards higher scores with advancing age (*p* < 0.001). 

The incidence of each obstetric complication is shown in Table 2. The proportion of almost all high-risk pregnancies were higher in older mothers. Late pregnancy complications were also higher in the AMA group. GDM treated with insulin and preeclampsia treated with magnesium sulfate were more frequently observed in the older age group (0.9% vs. 2.6% vs. 5.4%, 0.2% vs. 0.5% vs. 0.9%, respectively; *p* < 0.0001). Placental abnormalities including placental abruption and placenta previa also significantly increased with maternal age (0.4% vs. 0.6% vs. 0.6%, 0.9% vs. 2.1% vs. 3.4%, respectively; *p* < 0.0001). The rate of cesarean delivery was higher among the AMA group (34.8% vs. 46.0% vs. 59.7%; *p* < 0.0001).

Pregnancy complications associated with perinatal morbidity were more frequent in the AMA group. As the maternal age increased, the frequency of LBW and preterm birth also increased (1.5% vs. 2.8% vs. 3.9%, 1.6% vs. 2.3% vs. 3.0%, respectively; *p* < 0.0001). However, there was no significant increase in the incidence of stillbirth/IUFD. Postpartum complications including postpartum hemorrhage requiring transfusion, increased number of outpatient department (OPD) visits, and readmission were also associated with increasing maternal age. 

Comorbidities affecting pregnancy complications increase with an increase in maternal age. Therefore, we conducted logistic regression analysis after adjusting for the maternal demographics. Table 3 presents the aOR with 95% CIs for the assessment of the effect of increasing maternal age on each adverse pregnancy outcome, in comparison with young primigravida. Multivariate analysis demonstrated a substantial increase in the incidence of GDM/DM treated with insulin (age 35–39 years: aOR 2.04; 95% CI, 1.89–2.20 and age ≥ 40 years: aOR 3.21; 95% CI, 2.90–3.56), hypertensive disorder treated with magnesium during pregnancy (age 35–39 years: aOR 1.60; 95% CI, 1.34–1.90 and age ≥ 40 years: aOR 1.62; 95% CI, 1.23–2.13), and cesarean delivery (age 35–39 years: aOR 1.57, 95% CI, 1.54–1.59; age ≥ 40 years: aOR 2.59, 95% CI, 2.52–2.67). Pregnancy complications related to placental abnormalities, such as placenta previa (age 35–39 years: aOR 2.19, 95% CI, 2.07–2.32; age ≥ 40 years: aOR 3.66, 95% CI, 3.37–3.98) and placental abruption (age 35–39 years: aOR 1.29, 95% CI, 1.16–1.42; age ≥ 40 years: aOR 1.35, 95% CI, 1.12–1.63), were increased in the older age group.

AMA was also positively correlated with preterm birth (age 35–39 years: aOR 1.38, 95% CI, 1.32–1.46; age ≥ 40 years: aOR 1.70, 95% CI, 1.56–1.85) and LBW (age 35–39 years: aOR 1.71, 95% CI, 1.62–1.79; age ≥ 40 years: aOR 1.85, 95% CI, 1.71–2.00). AMA is a risk factor for postpartum complications after adjusting for variables. Older women were more likely to have readmission after childbirth (age 35–39 years: aOR 1.15, 95% CI, 1.08–1.24; age ≥ 40 years: aOR 1.27, 95% CI, 1.13–1.43). 

Before adjusting for maternal demographics, which may affect pregnancy outcomes, stillbirth/IUFD did not differ between the three groups. However, after adjustments, stillbirth/IUFD were more likely to occur in the older women, and this was statistically significant (age 35–39 years: aOR 1.37, 95% CI, 0.98–1.93; age ≥ 40 years: aOR 1.57, 95% CI, 0.87–2.84). 

Table 4 summarizes the relationship between maternal age and the number of additional medical interventions needed before and after delivery. Statistical analysis showed that variables such as the number of admissions before delivery and OPD visits after delivery were positively related to the maternal age, with more admissions and OPD visits required for older mothers.

Logistic regression for obstetric complications based on maternal age, using a threshold value of 35 years, is shown in Table 5.

## 4. Discussion

This study showed that AMA could be a risk factor for various types of adverse pregnancy outcomes, perinatal morbidity, and postpartum complications in South Korea. Most adverse pregnancy outcomes including GDM requiring insulin treatment, preeclampsia requiring magnesium sulfate administration, placenta previa, placental abruption, cesarean section, preterm birth, LBW, and postpartum hemorrhage were more frequently observed in women aged ≥35 years compared to those aged <35 years; moreover, the frequency of these adverse outcomes increased with an increase in maternal age. Our finding is consistent with those of the previous studies that have presented the association between AMA and adverse pregnancy outcomes. Since various maternal characteristics affect pregnancy outcomes, we adjusted for the maternal demographics and confirmed that AMA was still associated with poor pregnancy outcomes. 

It has been known that socioeconomic differences influence perinatal health such as preterm birth, LBW, and stillbirth [18]. Interestingly, the women in the AMA group who are in a better socioeconomic position showed poorer pregnancy outcomes than their younger counterparts. 

Obesity, DM, and hypertension are risk factors for GDM and preeclampsia. GDM and preeclampsia are one of the most common medical problems encountered during pregnancy, and the prevalence of those chronic diseases generally increases with maternal age [3,6,7,8]. Although the incidence of these comorbidities before pregnancy was low in this study, the prevalence of these diseases increased with age. In addition, a high prevalence of DM and hypertension are associated with other pregnancy complications, such as cesarean section, preterm birth, and LBW. 

This study evaluated the pre-pregnancy demographic characteristics of women in South Korea. Our data provide information about the lifestyle changes of Korean women with increase in age. In general, with increasing age, women earn more, live in the urban area, and are in a better socioeconomic position. Except for heavy alcohol drinking, the risk factors for adverse pregnancy outcomes, including smoking, obesity, DM, hypertension, and other medical comorbidities, were increased in the AMA population. These changes in demographics adversely affect pregnancy outcomes. It has been reported that smoking is associated with increased perinatal morbidity and stillbirth in all age groups. This risk is particularly high in older smokers [19,20]. The percentage of smokers increased with age in our study, which, to some degree, may contribute to a higher incidence of LBW and preterm labor in women of AMA.

Based on these observations, our result suggests that government policy supporting AMA pregnancy should focus not only on the economic support for the mother but also on the medical attention to maintain a healthy lifestyle for the mother.

The age of women at conception is the most important predictive factor for fertility. The prevalence of infertility increases with women’s age and women older than 35 years are recommended to receive an infertility evaluation and treatment after 6 months of failed attempts to conceive [21]. In addition, several studies have demonstrated that assisted reproductive technologies (ART) are associated with an increased risk of pregnancy complications [22,23]. 

Fitzpatrick KE et al. conducted a population-based cohort study using the UK obstetric surveillance system [24]. The authors reported that older women had more pregnancy complications including gestational hypertensive disorders, GDM, postpartum hemorrhage, cesarean delivery, and preterm birth. However, most effects of maternal age on pregnancy complications were attenuated or disappeared after adjustment for the mode of conception and multiple pregnancies. The authors suggested that ART is partly responsible for the increased risk of pregnancy complications. Our study included only primiparous women with singleton pregnancies; however, the subjects who underwent infertility treatment were also included, which is a confounding factor as indicated by Fitzpatrick KE et al. and may affect the relationship between maternal age and pregnancy. Kahveci B et al. examined the effect of AMA on perinatal outcomes with their Turkish cohort [13]. The authors excluded patients with a history of ART or multiple pregnancies in their study and still found the adverse effects of AMA on pregnancy outcome. Owing to contradictory results among studies, additional studies should be conducted to elucidate whether ART influences the outcomes in AMA pregnancy.

Our study presented a lower rate of prevalence in pregnancy complications compared with other similar studies examining the effect of AMA on pregnancy outcomes. In our study, the prevalence of GDM and preeclampsia was only 1.3% and 0.24%, respectively. This low prevalence is because we used the insurance claim data and extracted the data of subjects who received medications. In addition, our population characteristics may affect the result. Maternal ethnicity is reported to be associated with pregnancy-related morbidity and mortality. Dongarwar D et al. presented the difference in the prevalence of maternal–fetal outcomes, such as GDM and preeclampsia, in American women of AMA [25]. In the study, compared with non-Hispanic white women, Asian American women had reduced odds of GDM and preeclampsia. South Korea is an ethnically homogenous country with an absolute majority of Asian ethnicity. This ethnic difference may also result in a lower prevalence of GDM and preeclampsia. 

Older women have higher rates of spontaneous abortion in early pregnancy, and this risk of pregnancy loss associated with AMA has been well recognized. Most are attributed to fetal aneuploidy or fetal anomalies in the first trimester. However, our study did not include miscarriages from early pregnancies; if spontaneous abortions in early pregnancy were included, the risk of pregnancy complications associated with AMA would be higher.

A few limitations should be considered when interpreting the findings of this study. The diagnoses of maternal complications were based on insurance claims data from the NHIS database that was designed for cost claim issues and not for research purposes. Therefore, the main limitation is the validity of the diagnoses in this database. In addition, only those who participated in the NHSE were included in this study. Given that the participation rate in the NHSE was not substantially high and the subjects of this study had different characteristics, some limitations may be generalized to South Korea. Furthermore, data collected from the NHSE questionnaires are limited to the pre-pregnancy status. Therefore, it may not portray the effect of behaviors such as smoking, exercise, and alcohol consumption during pregnancy. Lastly, we could not identify those pregnant women who underwent assisted reproductive technology, which could impact pregnancy outcomes and was a confounding factor.

The strength of this study lies in its population-based cohort with a large sample size that included more than 500,000 primiparous women with almost no loss to follow-up. We utilized their data from the NHIS database to evaluate their hospitalization and OPD visits. The use of this large population-based cohort suggests that our findings may be widely generalized. In addition, to minimize and remove the effect of parity and history of cesarean delivery, we enrolled only primigravidae women. We also examined the medications used in the diagnosis for more accurate patient evaluation, rather than using the recorded diagnosis. In addition, unlike most other studies on the impact of maternal age, this is a contemporary, large retrospective study of primiparous women with singleton pregnancies, for whom the pre-pregnancy status could be evaluated. Most importantly, potential confounding factors in the relationship between maternal age and obstetric outcomes, including income level, BMI, smoking and alcohol consumption status, pre-existing medical conditions, area of residence, and exercise status were included.

## 5. Conclusions

As the maternal age at first pregnancy increases, there is an increase in the incidence of various obstetrical complications including preterm birth, GDM, preeclampsia, placental abnormalities, cesarean section rate, and postpartum complications requiring medical care. Therefore, information pertaining to obstetric outcomes associated with AMA should be provided accurately and promptly, and more vigilant surveillance is required in pregnancies involving AMA. To ensure a stable medical environment for clinicians and patients, social attention and government policy support are required for those of AMA at risk of pregnancy. 

## Figures and Tables

**Figure 1 jcm-11-00969-f001:**
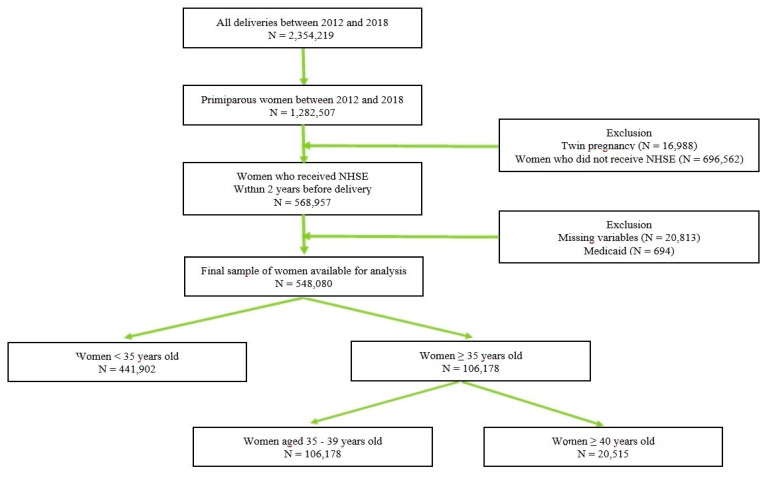
Study Flow Chart.

**Table 1 jcm-11-00969-t001:** Patients’ pre-pregnancy characteristics by maternal age.

Characteristic	Age <35 Years (*n* = 441,902)	Age 35–39 Years (*n* = 85,663)	Age ≥40 Years (*n* = 20,515)	*p*-Value
**Income level (%)**				**<0.001**
Low	12.0	11.0	15.5	
Low-middle	30.9	23.2	22.3	
Middle-high	41.8	41.5	31.6	
High	15.3	24.3	30.6	
**Residential area (%)**				**<0.001**
Capital	46.8	53.6	50.8	
Metropolitan	26.1	25.4	24.8	
Others	27.1	21.0	24.3	
**CCI (%)**				**<0.001**
0	69.5	66.8	61.7	
1	25.0	25.8	27.9	
2	5.6	7.5	10.5	
**Physically active (%)**	36.7	38.0	43.8	<0.001
**Current smoker (%)**	3.1	3.3	4.6	<0.001
**Heavy drinker (%)**	19.3	16.1	14.7	<0.001
**Pre-pregnancy** **BMI ≥ 25.0 kg/m^2^ (%)**	11.8	16.1	21.6	<0.001
**Pre-existing DM (%)**	0.2	0.4	1.0	<0.001
**Pre-existing hypertension (%)**	0.3	0.6	1.8	<0.001

Data are presented as %; CCI, Charlson Comorbidity Index; BMI, body mass index; DM, diabetes mellitus; Data are pre-pregnancy data, except for age; Chi-square tests were used to identify the differences in the distribution of general characteristics according to age group.

**Table 2 jcm-11-00969-t002:** Obstetric outcomes by maternal age.

Variables	Age < 35 Years (*n* = 441,902)	Age 35–39 Years (*n* = 85,663)	Age ≥ 40 Years (*n* = 20,515)	*p*-Value
**Late Pregnancy complication**				
Hospitalization *	62,602 (14.2)	14,233 (16.6)	3853 (18.8)	<0.0001
No. of hospitalizations	0.17 ± 0.48	0.21 ± 0.55	0.25 ± 0.60	<0.0001
Diabetes (with insulin)	3845 (0.9)	2210 (2.6)	1105 (5.4)	<0.0001
Preeclampsia (with magnesium)	716 (0.2)	458 (0.5)	183 (0.9)	<0.0001
Preterm labor with tocolytics	13,223 (3.0)	3668 (4.3)	1036 (5.0)	<0.0001
Cesarean delivery	153,662 (34.8)	39,409 (46.0)	12,256 (59.7)	<0.0001
Placenta previa	4129 (0.9)	1761 (2.1)	699 (3.4)	<0.0001
Placental abruption	191 (0.4)	485 (0.6)	19 (0.6)	<0.0001
**Perinatal morbidity**				
Preterm birth	7221 (1.6)	1989 (2.3)	607 (3.0)	<0.0001
Low birth weight	6606 (1.5)	2371 (2.8)	801 (3.9)	<0.0001
Stillbirth/IUFD	165 (0)	43 (0.1)	12 (0.1)	0.093
**Postpartum complication**				
Postpartum hemorrhage requiring transfusion	7135 (1.6)	2180 (2.5)	799 (3.9)	<0.0001
No. of OPD visits **	2.89 ± 2.17	3.01 ± 2.33	3.07 ± 2.48	<0.0001
Readmission ***	7135 (1.6)	2180 (2.5)	799 (3.9)	<0.0001

Data are presented as n (%) or mean ± standard deviation; OPD, outpatient department; IUFD, intrauterine fetal death; * before delivery, ** within 50 days of discharge, *** within 30 days of discharge; analysis of variance was used for No. of hospitalization/OPD visits and chi-squared test was used in the rest.

**Table 3 jcm-11-00969-t003:** Logistic regression for obstetric complications according to maternal age.

		Unadjusted OR (95% CI)		Adjusted OR (95% CI)	
Variables	Age < 35 y-o	Age 35–39 y-o	Age ≥ 40 y-o	Age 35–39 y-o	Age ≥ 40 y-o
**Late pregnancy complication**					
Hospitalization *	1.00	1.21 (1.18–1.23)	1.40 (1.35–1.45)	1.19 (1.17–1.21)	1.32 (1.27–1.37)
Preeclampsia (with magnesium)	1.00	2.02 (1.70–2.40)	3.00 (2.31–3.89)	1.60 (1.34–1.90)	1.62 (1.23–2.13)
Preterm labor with tocolytics **	1.00	1.45 (1.40–1.51)	1.72 (1.62–1.84)	1.39 (1.34–1.44)	1.47 (1.37–1.57)
Cesarean delivery	1.00	1.60 (1.58–1.62)	2.78 (2.71–2.86)	1.57 (1.54–1.59)	2.59 (2.52–2.67)
Placenta previa	1.00	2.33 (2.10–2.35)	3.74 (3.45–4.06)	2.19 (2.07–2.32)	3.66 (3.37–3.98)
Placental abruption	1.00	1.31 (1.18–1.45)	1.38 (1.15–1.65)	1.29 (1.16–1.42)	1.35 (1.12–1.63)
**Perinatal morbidity**					
Preterm birth	1.00	1.43 (1.36–1.51)	1.84 (1.69–2.00)	1.38 (1.32–1.46)	1.70 (1.56–1.85)
Low birth weight	1.00	1.71 (1.62–1.79)	1.85 (1.71–2.00)	1.71 (1.62–1.79)	1.85 (1.71–2.00)
Stillbirth/IUFD	1.00	1.35 (0.96–1.88)	1.57 (0.87–2.82)	1.37 (0.98–1.93)	1.57 (0.87–2.84)
**Postpartum complication**					
Postpartum hemorrhage requiring transfusion	1.00	1.60 (1.52–1.68)	2.44 (2.26–2.63)	1.59 (1.52–1.67)	2.47 (2.29–2.66)
Readmission ***	1.00	1.19 (1.11–1.27)	1.38 (1.22–1.55)	1.15 (1.08–1.24)	1.27 (1.13–1.43)

OR, odds ratio; CI, confidence interval; y-o, years old; IUFD, intrauterine fetal death; adjusted for history of conization, income level, residential area, Charlson Comorbidity Index, pre-pregnancy smoking, pre-pregnancy alcohol consumption, pre-pregnancy body mass index, history of diabetes, and history of hypertension. * before delivery, ** additionally adjusted for hypertensive disorder, *** within 30 days after discharge.

**Table 4 jcm-11-00969-t004:** Number of admissions and outpatient department visits.

Variables	Age < 35 Years	Age 35–39 Years		Age ≥ 40 Years		*p*-Value
		β	S.E.	β	S.E.	
No. of admissions before delivery	reference	0.1869	0.0093	0.2844	0.0167	<0.001
No. of OPD visits after delivery *	reference	0.0295	0.0022	0.0326	0.0041	<0.001

S.E, standard error; OPD, outpatient department. * Within 50 days of discharge.

**Table 5 jcm-11-00969-t005:** Obstetric outcomes by maternal age.

Variables	Age < 35 Years	Age ≥ 35 Years aOR (95% CI)	*p*-Value
**Late pregnancy complication**			
Hospitalization *	1.00	1.21 (1.19–1.24)	<0.0001
Diabetes (with insulin)	1.00	2.29 (2.15–2.45)	<0.0001
Preeclampsia (with Mg)	1.00	1.60 (1.36–1.88)	<0.0001
Preterm labor with tocolytics	1.00	1.40 (1.36–1.45)	<0.0001
Cesarean delivery	1.00	1.72 (1.70–1.75)	<0.0001
Placenta previa	1.00	2.47 (2.34–2.60)	<0.0001
Placental abruption	1.00	1.30 (1.18–1.43)	<0.0001
**Perinatal morbidity**			
Preterm birth	1.00	1.47 (1.38–1.51)	<0.0001
Low birth weight	1.00	1.74 (1.66–1.82)	<0.0001
Stillbirth/IUFD	1.00	1.41 (1.03–1.92)	0.030
**Postpartum complication**			
Postpartum hemorrhage requiring Transfusion	1.00	1.75 (1.68–1.83)	<0.0001
Readmission **	1.00	1.18 (1.11–1.25)	<0.0001

Data are presented as odds ratios (95% confidence interval) or β (standard error). CI, confidence interval; GDM, gestational diabetes mellitus; DM, diabetes mellitus, Mg, magnesium; IUGR, intrauterine growth restriction; IUFD, intrauterine fetal death. Adjusted for history of conization, income level, residential area, Charlson Comorbidity Index, smoking, drinking, exercise, body mass index, diabetes mellitus, and hypertension. * before delivery, ** within 30 days after discharge.

## Data Availability

Data cannot be shared publicly because of the protection of private information. Data belong to the National Health Insurance Service and are strictly controlled.

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
