# Peer review of "Impact of Maternal Age on Singleton Pregnancy Outcomes in Primiparous Women in South Korea"

_jcm, 2022, doi:10.3390/jcm11040969_

Round 1
Reviewer 1 Report
The manuscript “Impact of Maternal Age on Singleton Pregnancy Outcomes in Primiparous Women in South Korea” aimed to analysis the association between maternal age and pregnancy outcomes in singleton primigravidae.
Major concerns:
- Not only old age may have a negative impact on pregnancy outcomes, but the author should also analyze the outcomes with maternal age below 25 or 20 years.
- How many cases were conceived by ART?
- In table 1, the mean maternal age in the group of 35‒39 years was 36.3 and 41.4 in age >40 years. They were looked like not normal distribution Authors should use age as a continuous factor by multi-variants analysis to predict the outcomes. Arbitrarily divided age into below 35, 35-39 and over 40 may lose much information and not acceptable.
- In table 1, the incidence of poor factors like DM, H/T, smoking ….. were significantly different between groups by author’s definition, How to prove that is age, not those poor factors to influence the pregnancy outcomes. Authors should prove to us after deleting those confounding factors; age still is the determining factor to influence the pregnancy outcomes.
Author Response
Response to Reviewer 1 Comments
Major comments
Question 1. Not only old age may have a negative impact on pregnancy outcomes, but the author should also analyze the outcomes with maternal age below 25 or 20 years.
Response: We agree with the reviewer’s comment. The obstetric outcomes of mothers who are <25 years of age may differ from those of mothers who are > 25 years of age. When we designed the present study, we aimed to demonstrate the effects of elderly pregnancy on obstetric outcomes because our country has some social issues related to late marriage and elderly pregnancy. Therefore, our analysis focused on the obstetric outcomes of elderly pregnancy. Unfortunately, we could not also analyze the obstetric outcomes of younger women. We used the National health insurance service claim database. The government only provided the database for a specific period of time. For further analysis, another government approval would be required, and this will take a long time to acquire. However, as we mentioned above, we focused on elderly pregnancy and its impact on pregnancy outcome. In addition, important article on the impact of maternal age on obstetric outcomes also used the same age groups as those used in this study. Please see attached reference and consider it.
Question 2. How many cases were conceived by ART?
Response: We had tried to analyze the effects of ART on pregnancy outcomes. However, we could not identify only those women who underwent ART procedures and got pregnant. In addition, some women underwent ART procedures without receiving financial support from the National Health Insurance Service. Because we used the National Health Insurance Service claim data, we could not find those patients. Therefore, we did not include the ART as a confounding factor. We have also added this as one of the limitations of the present study in discussion section as follows;
“Lastly, we could not identify those pregnant women who underwent assisted reproductive technology, which could impact pregnancy outcomes and was a confounding factor.”
Question 3. In table 1, the mean maternal age in the group of 35‒39 years was 36.3 and 41.4 in age >40 years. They were looked like not normal distribution. Authors should use age as a continuous factor by multi-variants analysis to predict the outcomes. Arbitrarily divided age into below 35, 35-39 and over 40 may lose much information and not acceptable.
Response: We agreed with reviewer's comment. However, in many cohort study, the researchers divided the pregnant women into several groups according to their age. For example, Montori MG et al. (Taiwan J Obstet Gynecol. 2021 Jan;60 (1):119-124) conducted a cohort study using data from 27,455 singleton births at a single institution in Spain to examine the effect of advanced maternal age on pregnancy outcome. In their study, the authors showed that the data for maternal age did not meet the requirements for a normal distribution. Rademaker D. et al. (Acta Obstet Gynecol Scand. 2021 May;100(5):941-948) performed a cohort study to evaluate the association between adverse maternal and perinatal outcomes and very advanced maternal age. In their study, the authors used the Dutch perinatal registration and divided the women into age groups of 40–44, 45–49 and over 50 years of age. Cleary-Goldman et al. (Obstet Gynecol. 2005;105:983-990) also estimated the effect of maternal age on obstetric outcomes and divided the study participants into 3 age groups: 1) less than 35 years old; 2) 35–39 years old; and 3) 40 years of age and older. Many studies have also been conducted by grouping women according to their ages. Comparing the results of various studies is important for researchers; therefore, grouping women according to their age might be still effective.
Question 4. In table 1, the incidence of poor factors like DM, H/T, smoking ….. were significantly different between groups by author’s definition, How to prove that is age, not those poor factors to influence the pregnancy outcomes. Authors should prove to us after deleting those confounding factors; age still is the determining factor to influence the pregnancy outcomes.
Response: We agreed with reviewer’s comment. To eliminate confounding variables, we performed logistic regression analyses to estimate the adjusted odds ratio. The models were adjusted for all those variables including conization, income level, residential area, CCI, pre-pregnancy smoking, pre-pregnancy alcohol consumption, pre-pregnancy body mass index, history of diabetes, and history of hypertension.

Reviewer 2 Report
It is a well designed study with a large cohort of patients enrolled in it. The authors have managed to show that advanced maternal age is a risk factor for several morbidities and complications before and after gestation. I have one question do we know how many pregnant women were due to IVF ? It is a good study.
Author Response
Response to Reviewer 2 Comments
It is a well designed study with a large cohort of patients enrolled in it. The authors have managed to show that advanced maternal age is a risk factor for several morbidities and complications before and after gestation. I have one question do we know how many pregnant women were due to IVF ? It is a good study.
Response; Thanks for your comments. However, we could not identify only those women who underwent ART procedures including IVF and got pregnant. In addition, some women underwent ART procedures without the financial support of the National Health Insurance Service. Because we used the National Health Insurance Service claim data, we could not find those patients. Therefore, we did not include the ART as a confounding factor. We have also added this as one of the limitations of the present study in discussion section as follows;
Lastly, we could not identify those pregnant women who underwent assisted reproductive technology, which could impact pregnancy outcomes and was a confounding factor.
Round 2
Reviewer 1 Report
The authors do not respond to me well for the major concerns 3 and 4.
For question 3, the study group should with normal distribution so the statistic method authors used can be applied. In Table 1 and 2, every statistic result should point out which statistic method was used like the student t test or non-parameter test.
In table 3, DM with insulin use should not be a pregnancy outcome except as GDM; it should be a confounding factor
For question 4, Age should not as a determining factor for “Pregnancy Outcomes” if the woman is well without significant underlying factors. The author state “The models were adjusted for all those variables including conization, income level, residential area, CCI, pre-pregnancy smoking, pre-pregnancy alcohol consumption, pre-pregnancy body mass index, history of diabetes, and history of hypertension.” Authors should exclude those excluding those significant confounding factors then reanalysis.
Author Response
Response to review comments (Manuscript ID: jcm-1547696)
Dr. Emmanuel Andrès
Editor-in-Chief
Journal of Clinical Medicine
Dear Editor,
We are grateful for the opportunity to revise our manuscript for reconsideration for publication in the Journal of Clinical Medicine. We also deeply appreciate the reviewers for their helpful comments that have helped us to improve our manuscript and we are sorry for missing reviewer’s comments. We have carefully considered the reviewers’ comments and have made appropriate changes. We have tried our best to address all the comments.
Major comments
Question 3. The study group should with normal distribution so the statistic method authors used can be applied. In Table 1 and 2, every statistic result should point out which statistic method was used like the student- t test or non-parameter test.
In table 3, DM with insulin use should not be a pregnancy outcome except as GDM; it should be a confounding factor.
Response:
Thank you for your comments and we apologize for making confusion.
Data from categorical variables can not be properly analyzed using the statistical test based on the normal distribution. If we measured the age as an interval scale for the analysis, then we need to know whether the age is normally distributed or not. In this case, we still have a plan to apply a statistical test that is based on the normal distribution such as student t-test. However, the dependent variables that used for our study are categorical variables and the statistical tests that we adopted are distribution free tests. Please consider this.
Several historic articles in terms of the effects of maternal age on pregnancy outcome adopted similar approach that of ours.
In addition, the Central Limit Theorem states that sampling distribution of the sample means approach a normal distribution as the sample sized gets larger. When the population is normally distributed, it is known that the sample mean also follows a normal distribution, and by the ‘Central Limit Theorem’, even if the population does not follow a normal distribution, the above theorem holds similarly when there are enough data. I think we have enough population in our manuscript.
We categorized the age of pregnant women in three groups (< 35 years old, 35-39 years old, ≥ 40 years old)
Table 1 is comprised with categorical variables (income level, residential area, Charlson Comorbidity Index, physically active person (binary), Current smoker (binary), Heavy drinker (binary), Pre-pregnancy BMI (binary), Pre-existing DM (binary), Pre-existing hypertension (binary).
Thus, we used Chi-squared test in Table 1.
Table 2 is comprised with categorical and numeric variables
; Hospitalization (binary: yes or no), Diabetes (binary), Pre-eclampsia (binary), Preterm labor with tocolytics (binary), C-sec (binary), Placenta previa (binary), Placental abruption (binary), Preterm birth (binary),
Low birth weight (binary), Stillbirth (binary), Postpartum hemorrhage requiring transfusion (binary),
Readmission (binary) and two numeric variables (No. of OPD visits and No. of hospitalizations).
Thus, we used Chi-squared test for categorical variables and ANOVA for numeric variables in Table 2.
We revised it in footnote in each Table for better understanding of readers.
Additionally, we deleted the variable of ‘DM with insulin’ from confounding factor in Table 3 according to your comment.
Question 4. In table 1, the incidence of poor factors like DM, H/T, smoking ….. were significantly different between groups by author’s definition, How to prove that is age, not those poor factors to influence the pregnancy outcomes. Authors should prove to us after deleting those confounding factors; age still is the determining factor to influence the pregnancy outcomes.
Response: We are sorry for missing your point.
With age, pregnant women get many significant underlying factors including higher BMI, diabetes, hypertension and CCI, which could influence on adverse pregnancy outcomes.
According to your comment, we excluded confounding factors in main model and additionally we have suggested crude OR and 95% CI in Table 3.
Let us show unadjusted OR according to age group in Table 3.

This manuscript is a resubmission of an earlier submission. The following is a list of the peer review reports and author responses from that submission.